# Gastric Enzyme Supplementation Inhibits Food Allergy in a BALB/c Mouse Model

**DOI:** 10.3390/nu13030738

**Published:** 2021-02-26

**Authors:** Nazanin Samadi, Denise Heiden, Martina Klems, Martina Salzmann, Johanna Rohrhofer, Eleonore Weidmann, Larissa Koidl, Erika Jensen-Jarolim, Eva Untersmayr

**Affiliations:** 1Institute of Pathophysiology and Allergy Research, Center of Pathophysiology, Infectiology and Immunology, Medical University of Vienna, 1090 Vienna, Austria; nazanin.samadi@meduniwien.ac.at (N.S.); denise.heiden@meduniwien.ac.at (D.H.); martina.klems@meduniwien.ac.at (M.K.); martina.salzmann@meduniwien.ac.at (M.S.); johanna.rohrhofer@meduniwien.ac.at (J.R.); eleonore.weidmann@gmx.at (E.W.); larissa.koidl@meduniwien.ac.at (L.K.); erika.jensen-jarolim@meduniwien.ac.at (E.J.-J.); 2The Interuniversity Messerli Research Institute of the University of Veterinary Medicine Vienna, Medical University of Vienna, and University of Vienna, 1210 Vienna, Austria

**Keywords:** food allergy, gastric enzymes, gastric acid suppression, systemic allergic response, allergy prevention

## Abstract

Impaired gastric digestion due to suppressed gastric acidity enhances the risk for food allergy development. In the current study, we aimed to evaluate the impact of a supported gastric digestion via application of a pharmaceutical gastric enzyme solution (GES) on food allergy development and allergic reactions in a BALB/c mouse model. The ability of the GES to restore hypoacidic conditions was tested in mice treated with gastric acid suppression medication. To evaluate the impact on allergic symptoms, mice were orally sensitized with ovalbumin (OVA) under gastric acid suppression and subjected to oral challenges with or without GES. The immune response was evaluated by measurement of antibody titers, cytokine levels, mucosal allergy effector cell influx and regulatory T-cell counts. Clinical response was objectified by core body temperature measurements after oral OVA challenge. Supplementation of GES transiently restored physiological pH levels in the stomach after pharmaceutical gastric acid suppression. During oral sensitization, supplementation of gastric enzymes significantly reduced systemic IgE, IgG1 and IgG2a levels and allergic symptoms. In food allergic mice, clinical symptoms were reduced by co-administration of the gastric enzyme solution. Support of gastric digestion efficiently prevents food allergy induction and alleviates clinical symptoms in our food allergy model.

## 1. Introduction

It is well established that stability of food proteins to gastrointestinal digestion affects protein allergenicity. Higher resistance to digestion along the digestive tract enhances the sensitization capacity of food proteins and supports the elicitation of an allergic response [1]. Under physiological conditions, food proteins enter the stomach and are denatured by acidic gastric fluids to facilitate enzymatic degradation. Moreover, low intragastric pH levels are essential for gastric enzyme activation, with pepsin being one of the key enzymes to initiate food protein digestion along the gastrointestinal tract. Gastric pH varies according to age and stomach content [2]. In a fasted state, the pH of gastric fluids is between 1.5 and 3 in healthy individuals [3,4]. This may vary substantially, ranging from pH 1.0 to pH 8.0 even in healthy individuals, with short time periods of high intragastric pH levels due to swallowing of water or saliva or due to reflux of duodenal fluids [2]. The main gastric protease pepsin is released into the gastric lumen as the inactive pro-enzyme pepsinogen [5]. Upon exposure to the acidic environment, the electrostatic interaction between pepsin and the pro-segment is disrupted, resulting in release of the active protease pepsin [5]. In a fasted state, pepsin concentrations vary between 0.11 and 0.22 mg/mL [6]. The amount of pepsin is highly individual and also depends on the type of food ingested [7]. After eating, pepsin production increases and concentrations of up to 0.58 mg/mL are reached.

In repeated in vitro experiments, we were able to demonstrate that a proper digestion of important food proteins such as hazelnut, codfish, egg and milk is inhibited when the pH of digestive fluids is elevated [8,9,10,11]. Additionally, murine as well as human studies confirmed the association between hypoacidic gastric pH due to medication with gastric acid suppressive drugs and food allergy development [8,9,10,12,13]. These data were confirmed by other groups reporting food allergy development after intake of anti-ulcer drugs such as proton pump inhibitors (PPIs) also in the pediatric population in large cohort studies [14,15,16]. Moreover, in a study with fish allergic patients we confirmed the protective role of gastric digestion in an already-established food allergy [17]. Anti-ulcer drugs are applied for treatment of dyspeptic disorders of the gastrointestinal tract [18]. These drugs elevate the gastric pH by either blocking H2 receptors or proton pumps of gastric parietal cells, which leads to reduced or completely abrogated acid secretion [5]. However, due to several studies reporting side-effects of gastric acid suppression medication, PPIs are no longer considered as harmless, especially in situations of long-term drug application [19]. Very recently, a population-wide study confirmed the correlation between gastric acid suppression and allergy. A highly significant association between prescription of gastric acid suppression medication and anti-allergy drugs was detected over time [20], confirming again the allergy-protective function of gastric digestion.

Based on these data, we aimed to advance this already-established knowledge in the present study. To evaluate the impact of enzyme supplementation on food allergy development and on food allergic reactions, we administered a gastric enzyme solution (GES) orally to support gastric digestion in our established BALB/c food allergy mouse model [21]. The GES contains the main proteolytic gastric enzyme, pepsin, together with amino acid-hydrochloride, aiming to decrease the luminal gastric pH and to induce an environment favorable for the protease activity. This study tested our hypothesis that support of gastric digestion might interfere with food allergy development and might protect against food allergic reactions.

## 2. Materials and Methods

### 2.1. Animals

For all experiments, 94 female BALB/cAnNCrl mice (15–20 g, provided with a health report certificate) were purchased from the Center of Biomedical Research, Medical University of Vienna, Division for Laboratory Animal Science and Genetics (Himberg, Austria) and housed under conventional conditions. Mice were randomly divided into groups and had access to food (egg-protein-free diet, Ssniff, Soest, Germany) and water ad libitum. Mice were treated according to the European Union guidelines of animal care and with permission from the animal ethics committee of the Medical University of Vienna and the Austrian Federal Ministry of Science and Research (permission numbers: BMWFW-66.009/0036-WF/V/3b/2016 and BMWFW-66.009/0042-V/3b/2019).

### 2.2. Mouse Treatment Protocols

To evaluate the effect of pharmaceutical gastric enzymes (Enzynorm^®^; Nordmark Pharma GmbH, Uetersen, Germany; containing cathepsin, pepsin (8.75 pepsin units per tablet) and protein-bound hydrochloric acid) in situations of reduced gastric acid production, mice (experiment 1: 5 groups, *n* = 6 each group) were injected with the proton pump inhibitor (PPI) omeprazole (OMEP^®^ Hexal, Holzkirchen, Germany) on 14 consecutive days (Appendix A). The dosage of the gastric enzymes that is used for supplementation in humans was adjusted to the mouse body weight as well as to the accelerated rodent metabolism [22]. Each dosage contained 500 µg or 1000 µg of pepsin and was administered in solubilized form (GES). Intragastric pH was measured by aspiration of gastric fluid immediately or 5 min after gavage. For food allergy induction (experiment 2), mice of groups A and B (*n* = 10 each group) were gastric acid suppressed by injection of 116 μg of omeprazole dissolved in 0.9% sodium chloride on 3 consecutive days every second week (Appendix A). On days 2 and 3 of each of the 5 immunization cycles, mice were fed with 200 μg of ovalbumin (OVA; Sigma Aldrich, St. Louis, MO, USA) mixed with the GES containing Enzynorm^®^ with 1000 µg of pepsin (group A) or 200 μg of OVA (group B). For both mouse groups, sucralfate (Sucralan^®^ Gerot Lannach Pharma Company, Lannach, Austria) was added. Group C served as positive control regarding IgE induction and was injected with 2 µg of OVA adsorbed to aluminum hydroxide as an adjuvant. Mice of group D remained naïve throughout the immunization period. After allergy induction, mice were orally challenged with 2 mg of OVA and the core body temperature was measured before and 15, 30, 45 and 60 min after oral challenge. Immediately after final anesthesia, blood was collected by cardiac puncture. Spleens and intestinal lavages were harvested for further evaluations. From each animal’s gastric antrum, tissue was collected for histological evaluations.

For food allergy induction, mice (experiment 3: 4 groups, *n* = 8 each group) were gastric acid suppressed by injection of 116 μg of omeprazole dissolved in 0.9% sodium chloride on 3 consecutive days every second week (Appendix A). On days 2 and 3 of each of the 6 immunization cycles, mice were fed with 200 μg of OVA (Sigma Aldrich, St. Louis, MO, USA) mixed with sucralfate. Afterwards, mice were orally challenged with 2 mg of OVA. The first group received 1000 μg of GES 5 min before oral challenge, the second group was challenged with a mixture of OVA and 1000 μg of GES and the third group only received the oral OVA challenge. The allergic response was evaluated by core body temperature measurements.

### 2.3. Detection of OVA-Specific Antibodies in Sera and Intestinal Lavages

Blood was collected from the facial vein or via cardiac puncture after final read-out experiments. Small intestines were removed and flushed with 2 mL of PBS with protease inhibitor (Complete Mini, Roche, Basel, Switzerland). Serum samples were screened for OVA-specific IgE, IgG1 and IgG2a and lavages for total and OVA-specific IgA by enzyme-linked immunosorbent assay (ELISA) as previously described [23].

### 2.4. Rat-Basophil Leukemia Cell (RBL) Assay

In order to analyze the biological activity of OVA-specific IgE antibodies, a rat-basophil leukemia cell assay (RBL assay using the RBL-2H3 cells, a kind gift of Arnulf Hartl, Paracelsus Medical University, Salzburg, Austria) was performed as previously described [24].

### 2.5. Hematoxylin and Eosin Staining

On the day of final read-out experiments, gastric antrum tissue of each mouse was dissected, embedded in Tissue TEK^®^ O.C.T. compound (Sanova Pharma, Vienna, Austria) and stored at −80 °C. Cryosections (4 µm) were prepared with a Cryostat (Cryostat Leica CM3050, Leica Biosystems, Wetzlar, Germany) and stored at −20 °C. For the staining procedure, tissue sections were washed after defrosting. Hematoxylin (1:1 in distilled water) was added for 5 min and subsequently washed away. Eosin Y (+ 0.05% acetic acid) was applied for 1 min and again washed away. Dehydration was achieved by an ascending ethanol series. Samples were incubated two times in N-butylacetate (Sigma Aldrich) and covered with Eukitt^®^ (Sigma Aldrich) and glass cover slips to preserve the tissue. Stained sections were acquired with TissueFAXS (TissueFAXS Version 4.2.6245.1019, TissueGnostics GmbH, Vienna, Austria) and analyzed with HistoQuest (HistoQuest Version 6.0.1.125, TissueGnostics GmbH). Five areas of the tunica mucosa were selected randomly and screened per sample to evaluate existence of tissue inflammation. Accumulation of inflammatory cells was evaluated and numbers of eosinophils were normalized to area size.

### 2.6. Toluidine Staining

For unfreezing, samples were kept for 30 min at room temperature. Afterwards, the slides were washed with PBST (0.1% Tween) for 30 min followed by a 5 min washing step with distilled water. A 0.1% toluidine blue solution (Sigma Aldrich) was applied for 5 min to each sample. After two more washing steps with distilled water, dehydration was achieved by an ascending ethanol series followed by two final incubation steps in N-butylacetate (Sigma Aldrich) for 5 min each. To preserve the tissue, samples were mounted with Eukitt^®^ (Sigma Aldrich) and covered with glass cover slips (Thermo Fisher Scientific, Waltham, MA, USA). Stained sections were acquired with TissueFAXS (TissueFAXS Version 4.2.6245.1019, TissueGnostics GmbH) and analyzed with HistoQuest (HistoQuest Version 6.0.1.125, TissueGnostics GmbH). Five areas of the tunica mucosa were selected randomly and screened per sample to evaluate existence of tissue inflammation. Numbers of mast cells were normalized to area size.

### 2.7. Statistics

The generated data were statistically compared using GraphPad Prism version 5.00 for Windows (GraphPad Software, San Diego, CA, USA). All results were checked for normal distribution by the Kolmogorov–Smirnov test. Intragastric pH measurement, IgE, IgG1, Ig2a levels, body core temperature as well as inflammatory cells were analyzed with the Kruskal–Wallis non-parametric test and Dunn’s multiple correction. Comparison of body temperature changes, total and specific IgA and RBL assay were compared with ANOVA combined with Tukey’s post-test. A *p* value < 0.05 was considered as statistically significant.

## 3. Results

### 3.1. Gastric Enzyme Solution Lowers Elevated Gastric pH in a Dose-Dependent Manner

Our first experimental approach was to confirm the effect of GES on gastric pH levels (Appendix A). Mice received PPIs as a gastric acid suppression medication daily for 2 weeks. Intragastric pH levels were measured immediately or 5 min after gavage of different dosages of GES containing 500 µg or 1000 µg of pepsin. Our results indicate that a single administration was only transiently able to significantly decrease the hypoacidic intragastric pH levels (Figure 1).

### 3.2. Prevention of OVA-Specific Antibody Formation upon Sensitization under Gastric Acid Suppression by Concomitant Administration of Gastric Enzyme Solution

In a second experiment, we assessed the impact of GES on food allergy development in our food allergy mouse model with repeated oral sensitizations under concomitant anti-ulcer medication with omeprazole and sucralfate [21] (Appendix A). Mice were orally sensitized under gastric acid suppression with or without GES or injected with the allergen intraperitoneally (ip) as positive controls for IgE induction. Of interest, 6 out of 10 animals receiving oral immunizations under gastric acid suppression had elevated IgE levels, while IgE levels remained at background levels in the animals orally immunized under anti-ulcer drugs but supplemented with GES (Figure 2a). These results were confirmed by RBL assays. While cells incubated with sera from allergic mice immunized orally under gastric acid suppression or injected ip with OVA showed a significantly higher mediator release, passive sensitization of mast cells with sera from animals orally sensitized under concomitant GES administration revealed no mediator release (Figure 2b). Similar results were observed for OVA-specific IgG1 titers (Figure 2c). OVA-specific IgG2a remained at the baseline after sensitizations in both groups sensitized via the oral route (Figure 2d). Total and OVA-specific IgA levels measured in intestinal lavages were comparable in all groups (Appendix A).

### 3.3. No Histological Signs of Inflammation Were Found in Gastric Antrum Mucosa

To evaluate the safety of enzyme supplementation and the impact on local inflammatory cells (mast cells and eosinophils), we screened for allergy effector cells in the gastric mucosa. The numbers of the cells were similar in sensitized mice receiving GES and naïve mice, indicating a preserved mucosal homeostasis (Figure 3).

### 3.4. Negligible Effect of GES on Regulatory T Cells and Cytokine Levels

Regulatory T cells (Tregs) were isolated from splenocytes and characterized by flow cytometry. Despite the different immunization protocols, the number of Tregs was comparable to the cell count detected in the naïve group. Moreover, splenocytes were stimulated with OVA (5 µg/mL) and the Th2, Treg and Th1 cytokine levels (IL-4, IL-10 and IFN-γ) were measured in supernatants by cytokine ELISA. There was no variation regarding immunization strategies and only baseline levels were detected in all groups except for mice after ip treatment showing significantly elevated levels for all three cytokines.

### 3.5. Protective Effect of GES against Systemic Allergic Response upon Oral Challenge

To evaluate the clinical response in allergic mice, oral provocations with OVA were performed on the day of final read-out experiments and the core body temperature was recorded for one hour after oral challenges to evaluate allergic reactions. The group sensitized orally with PPIs and GES was completely protected against allergic reactions, while both other sensitized groups (oral OVA administration under gastric acid medication and OVA ip injections) showed a drop of core body temperature indicating an allergic response (Figure 4).

### 3.6. Protection against Allergic Reactions upon Oral Allergen Provocation by Concomitant Administration of GES

In a third experiment, the effect of GES on allergic symptoms was evaluated in allergic mice after six oral immunization cycles (Appendix A). OVA allergic mice with comparable OVA-specific serum IgE, IgG1, IgG2a and mucosal IgA titers (Appendix A) were subjected to oral OVA provocations. Core body temperature was measured to objectify an allergic response. Co-administration of GES with OVA resulted in significantly reduced allergic reactions indicated by only a minor temperature drop after 10 min (Figure 5), 30 min and 60 min. The allergic mice receiving GES 5 min before oral challenge or no GES supplementation were not protected against an allergic response indicated by a drop of core body temperature (Figure 5).

## 4. Discussion

As food allergies present an increasing health concern in western society, affecting up to 10% of the general population, it is pivotal to define and develop efficient allergy prevention measures. To highlight the importance of this topic, food allergy is not only the main cause for severe, even life-threatening, allergic reactions [25]. The tremendous personal and public health burden of food allergies also drives the worldwide search for commonly accessible food allergy prevention strategies. Thus, it is essential to understand the mechanism associated with disease development. As indicated above, we have repeatedly confirmed the association between medication with gastric acid suppressive drugs and allergy development [5,9,10,20]. In recent years, concerns have been raised associated with the over-the-counter availability of these drugs and the frequent long-term use of patients without seeking medical advice or a proper diagnosis [19,26,27]. Besides the postulated impact on protein digestion, PPIs are also known to influence microbiota composition [28]. Especially in the gastrointestinal tract, commensal microbes are recognized for their substantial protective influence on food allergy development [29]. In accordance, recent studies linked a change in intestinal microbiome composition and function to several chronic inflammatory diseases affecting different metabolic pathways including non-alcoholic fatty liver disease, celiac disease and type 1 diabetes mellitus [30,31,32]. Of interest, we have previously demonstrated that also during sensitization under concomitant gastric acid suppression, the microbiota composition confers a protective effect against food allergy development [33]. Thus, gastric acid suppression might impact the immune response via different mechanisms. However, it should not be forgotten that anti-ulcer drugs are also essential treatment options for dyspeptic disorders, which have tremendously decreased the need for surgical treatment of dyspepsia since their marketing in the 1970s [18]. Therefore, it is of high medical interest to develop strategies allowing the use of this important medication without the potential side effects associated with allergy development.

To the best of our knowledge, this is the first study evaluating the support of gastric digestion for protection against food allergy development. In our study, we were able to demonstrate that sensitization under gastric enzyme supplementation resulted in reduced formation of allergen-specific antibody titers and protection from an allergic response upon allergen challenge. Thus, our data open new avenues for protection from food allergy development in situations with reduced gastric acid production such as treatment with anti-ulcer medication for dyspeptic diseases.

Despite major research efforts on immune modulating treatment options [34], strict allergen avoidance still remains the primary solution for food allergic patients to avoid severe allergic reactions. Accidental allergen ingestion is a major problem, especially in younger food allergy patients [35]. Therefore, new strategies are essential to enhance patients’ safety in case of inadvertent food allergen exposure. In our study, we have demonstrated protection against allergic symptoms as indicated by a core body temperature drop in animals receiving support of gastric digestion via administration of gastric enzymes. Of interest, the effect seemed to be transient, as gastric enzymes had to be co-administrated with the food allergens to induce the allergy protective effect.

We are fully aware of the limitations of our study. With regards to investigation of immunological mechanisms, there are substantial differences in immune cell function between human patients and mouse models [36]. Nevertheless, our mouse model was standardized in repeated experiments [21] and reflects the situation in human patients [9,10]. Moreover, detailed kinetic experiments will be helpful to define the protective time after ingestion of the gastric enzyme tablet. Without any doubt, the data generated in a mouse model cannot be directly extrapolated to human patients due to differences in metabolism [22]. Moreover, there might be differences regarding age groups. Clinical pediatric guidelines suggest introduction of allergenic foods in the first year of life to improve oral tolerance and prevent development of food allergies [37,38]. With the emerging knowledge on protein stability and integrity during digestion, this approach raises questions on food design for a safe diet and improved health management in infants, as children show a distinct gastric function compared to adults [39,40,41]. Based on this information, given the pre-clinical data presented in this study and given that the medication used in this study is already approved for patients, we suggest a rapid transition into a clinical setting for evaluation of the allergy protective efficacy of gastric enzymes.

In conclusion, supplementation of gastric enzymes via a pharmaceutical preparation was shown to transiently restore the physiological pH levels in the stomach after gastric acid suppression. Supporting gastric digestion by enzyme supplementation was associated with protection against food allergy development evidenced by decreased systemic allergen-specific antibody titers, no drop of body temperature after oral allergen challenge and no influx of inflammatory cells into the gastric mucosa. Additionally, allergic mice were protected against allergic reaction by gastric enzyme co-administration. Our findings clearly demonstrate an allergy preventive effect of gastric enzymes when administered together with food proteins. Supporting gastric digestion might, thus, prevent food allergy development and might even inhibit systemic allergic reactions in already established food allergy. Given the substantial impact of food allergy on the quality of life of affected patients as well as the associated nutritional limitation [42,43], our study provides first evidence for future prevention strategies, which will substantially support affected patients.

## Figures and Tables

**Figure 1 nutrients-13-00738-f001:**
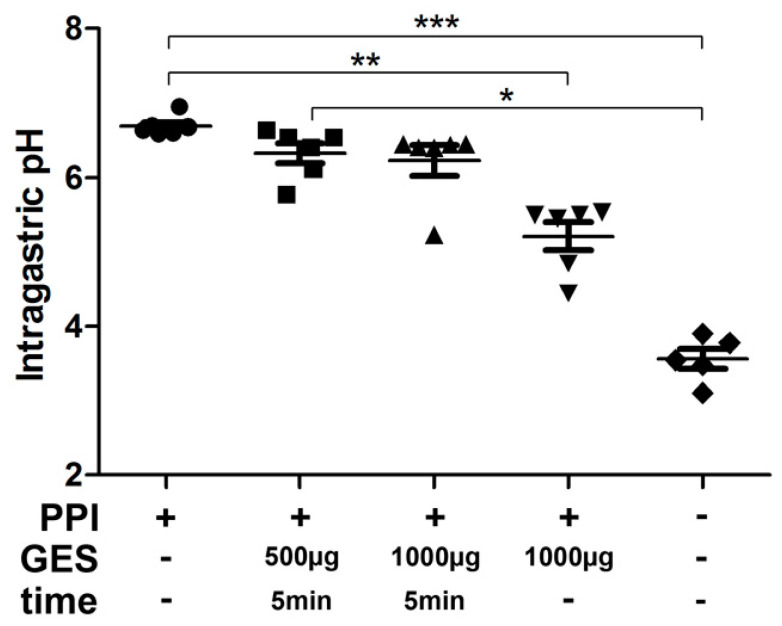
Gastric hypoacidic pH levels are transiently restored after administration of a gastric enzyme solution. After 2 weeks of proton pump inhibitor (PPI) treatment, intragastric pH values were evaluated. One group of mice received no further treatment before measurement. All other measurements were done 5 min after administration of 500 μg of gastric enzyme solution (GES), 5 min after gavage of 1000 μg of GES, immediately after administration of 1000 μg of GES or in untreated, naïve animals. PPI: gastric acid suppressive treatment. GES: amount of GES administration. Time: time interval between GES administration and intragastric pH measurement. * *p* < 0.05, ** *p* < 0.01, *** *p* < 0.001.

**Figure 2 nutrients-13-00738-f002:**
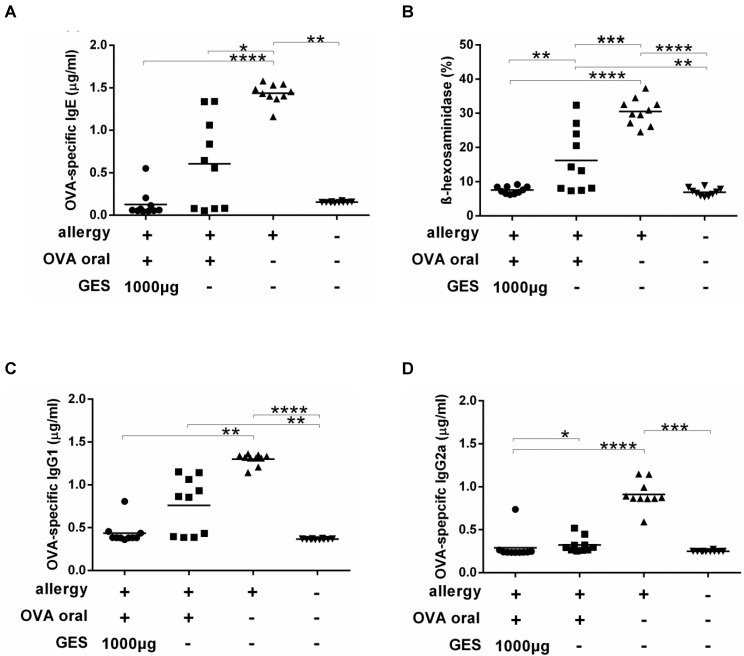
Oral sensitization under supplementation with gastric enzyme solution (GES) is associated with suppressed systemic allergen-specific antibody formation. Blood was collected for antibody detection after oral immunizations with ovalbumin (OVA) under gastric acid suppression with or without concomitant GES after intraperitoneal (ip) injections of OVA as the positive control group or from naïve animals as negative controls. OVA-specific IgE (**A**), IgG2a (**C**) and IgG1 (**D**) antibodies were measured and the functionality of OVA-specific IgE was indicated by rat-basophil leukemia cell (RBL) assay (**B**). (* *p* < 0.05, ** *p* < 0.01, *** *p* < 0.001, **** *p* < 0.0001).

**Figure 3 nutrients-13-00738-f003:**
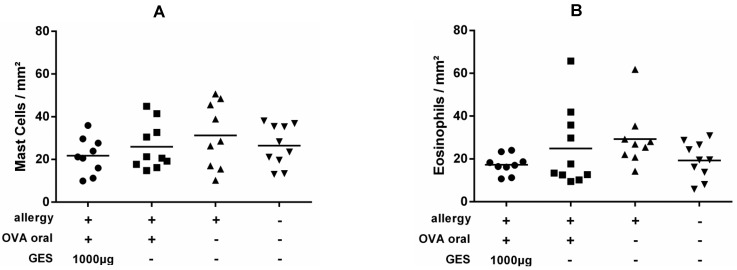
Evaluation of inflammatory cells in antrum tissue after food allergy induction. The number of mast cells (**A**) and eosinophils (**B**) were normalized to the size of the evaluated tissue. Results from groups receiving oral ovalbumin (OVA) sensitizations under gastric acid suppression with and without gastric enzyme solution (GES) supplementation, the positive controls after ip immunizations and the naïve animals were compared by the Kruskal–Wallis non-parametric test and Dunn’s multiple correction tests.

**Figure 4 nutrients-13-00738-f004:**
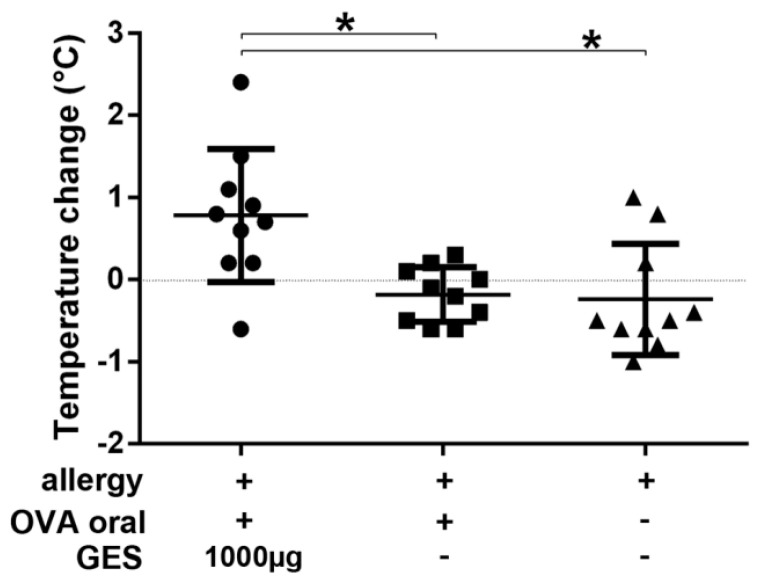
Gastric enzymes protect against clinical reactivity after oral provocation with ovalbumin (OVA). Evaluation of core body temperature after oral OVA challenge in OVA allergic mice. Mice were sensitized orally with OVA under gastric acid suppression with (left group) or without concomitant gastric enzyme solution (GES) supplementation (middle group) or were injected with OVA ip as positive controls (right group). Dots indicate the temperature differences for each mouse from before to 15 min after oral challenge. (* *p* < 0.05).

**Figure 5 nutrients-13-00738-f005:**
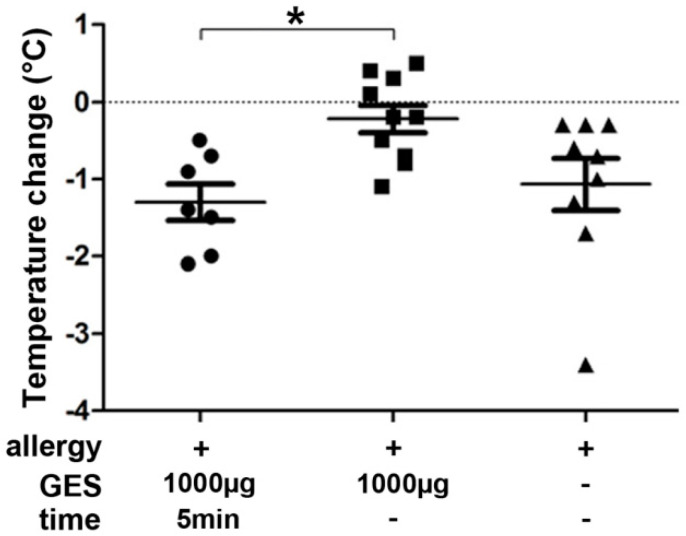
Gastric enzymes protect allergic mice against anaphylaxis after oral provocation with OVA. Evaluation of core body temperature after oral OVA challenge in OVA allergic mice. Dots indicate the temperature differences for each mouse from before to 15 min after oral challenge. OVA allergic mice were challenged with 2 mg of OVA and received 1000 μg of gastric enzyme solution (GES) 5 min before oral challenge with OVA (left group), 1000 μg gastric enzymes together with the OVA challenge (middle group) or the allergen alone (right group). (* *p* < 0.05).

## Data Availability

The data presented in this study are available in the article and Appendix A. Cytokine data reported in this study are available on request from the corresponding author. These data are not publicly available due to limited impact on study results.

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
