# Peer review of "Gastric Enzyme Supplementation Inhibits Food Allergy in a BALB/c Mouse Model"

_nutrients, 2021, doi:10.3390/nu13030738_

Round 1
Reviewer 1 Report
I encourage the authors to extend the references regarding the role of microbiota in health and disease when they mention that “especially in the gastrointestinal tract, commensal microbes are recognized for their substantial protective influence on food allergy development”. I suggest to add these references with regard to other chronic conditions: PMID: 25400436, PMID: 31597349 and PMID: 31684011
Author Response
Comment 1: I encourage the authors to extend the references regarding the role of microbiota in health and disease when they mention that “especially in the gastrointestinal tract, commensal microbes are recognized for their substantial protective influence on food allergy development”. I suggest to add these references with regard to other chronic conditions: PMID: 25400436, PMID: 31597349 and PMID: 31684011
Response 1: We are very thankful for this important comment. Based on the reviewer’s suggestion, we have amended the references in order to highlight the change in microbiome composition and function in situations of chronic inflammatory diseases affecting different pathways of the human metabolism (line 293 – 296)

Reviewer 2 Report
I read with pleasure this study guided by a clear hypothesis, confirmed by the experiments described. The article is clearly and well written. The introduction, although a little redundant, filters the conclusions. The methods are clearly described and all data are discussed. The discussion could probably be reduced with the abolition of its first part: it really does not discuss the exposed data, but is narrative and repetitive of the introduction.
In general, the thesis formulated is clearly documented and a suggestion of possible interest for both the diagnosis and the treatment of food allergy arises. If the proposed hypothesis will be confirmed, the results of this study are counterintuitive to the dominant theory that in order to reduce food allergy, food proteins should be given as early and abundantly as possible, thus exposing children to intact food allergens at an age when their peptic digestion is not yet highly efficient. The authors should be prepared for such objections. They could consider inserting a comment like this, perhaps corroborated by other data documenting how weakening digestive barriers increases food allergy in different models. I am referring to clinical data on milk allergy in short bowel syndrome.
Specific.
Figure 1 – the line ‘time’ deserves more detail: it is not immediately intuitive by looking at the figure and reading the caption what the timing of the experiments is.
Author Response
Comment 1: I read with pleasure this study guided by a clear hypothesis, confirmed by the experiments described. The article is clearly and well written. The introduction, although a little redundant, filters the conclusions. The methods are clearly described and all data are discussed. The discussion could probably be reduced with the abolition of its first part: it really does not discuss the exposed data, but is narrative and repetitive of the introduction.
Response 1: We are very thankful for the favourable review and the reviewer’s comments on our manuscript. The redundant text passages of the discussion have been deleted (line 270 – 277 and line 286 - 287).
Comment 2: In general, the thesis formulated is clearly documented and a suggestion of possible interest for both the diagnosis and the treatment of food allergy arises. If the proposed hypothesis will be confirmed, the results of this study are counterintuitive to the dominant theory that in order to reduce food allergy, food proteins should be given as early and abundantly as possible, thus exposing children to intact food allergens at an age when their peptic digestion is not yet highly efficient. The authors should be prepared for such objections. They could consider inserting a comment like this, perhaps corroborated by other data documenting how weakening digestive barriers increases food allergy in different models. I am referring to clinical data on milk allergy in short bowel syndrome.
Response 2: The reviewer raises a very interesting point. As clinical guidelines suggest an early introduction of allergens in infants to improve oral tolerance, a special focus on protein stability and pediatric gastric functionality is essential. According statements have been included in the discussion (line 329 - 334).
Specific:
Comment 3: Figure 1 – the line ‘time’ deserves more detail: it is not immediately intuitive by looking at the figure and reading the caption what the timing of the experiments is.
Response 3: We agree with the reviewer that the figure caption would benefit from a refinement. We have adjusted the figure caption to make the treatment and timeline clearer to the reader (lines 187-188).
